# Automated Complete Blood Cell Count Using Sysmex XN-9000^®^ in the Diagnosis of Newborn Infection

**DOI:** 10.3390/jcm11195507

**Published:** 2022-09-20

**Authors:** Nils Wettin, Tim Drogies, Andreas Kühnapfel, Berend Isermann, Ulrich Herbert Thome

**Affiliations:** 1University Hospital for Anesthesiology, Martin Luther University of Halle-Wittenberg, Ernst-Grube-Straße 40, 06120 Halle, Germany; 2Insitute for Clinical Chemistry, University Hospital, Paul List Straße 13-15, 04103 Leipzig, Germany; 3Institute for Medical Informatics, Statistics and Epidemiology, Medical Faculty of the University, Härtelstraße 16-18, 04107 Leipzig, Germany; 4Division of Neonatology, University Hospital for Children, Liebigstraße 20a, 04103 Leipzig, Germany

**Keywords:** complete blood cell count, neonatal sepsis, Sysmex, immature granulocytes

## Abstract

The early identification of septically infected newborn infants is important for ensuring good outcomes. Blood cell differentiations are helpful, but they are often time consuming and inaccurate. In this study, we evaluated the use of automatic white blood cell differentiations by flow cytometry for the diagnosis of neonatal sepsis. Episodes of suspected infection in neonates were retrospectively classified into two groups, unlikely infection (UI, levels of Interleukin-6 < 400 pg/mL or CRP within 48 h < 10 mg/L), *n* = 101 and probable infection (PI, Interleukin-6 ≥ 400 pg/mL or CRP within 48 h ≥ 10 mg/L), *n* = 98. Complete blood cell counts were performed by Sysmex XN-9000^®^ using flow cytometry. Relative and absolute proportions of immature granulocytes were evaluated. Unexpectedly, the absolute count of immature granulocytes was significantly lower in the group of PI compared to UI neonates. Similar results were found when analysing the relative proportion of immature granulocytes among all neutrophil granulocytes. On the other hand, manually counted immature to total (I/T) ratios of granulocytes were higher in PI than in UI infants. Therefore, we conclude that differentiations of granulocytes by Sysmex XN-9000^®^ can be used to distinguish between infected and uninfected neonates if the results are interpreted according to our findings. A low count of immature granulocytes as determined by Sysmex XN-9000^®^ may indicate neonatal infection.

## 1. Introduction

Early detection of neonatal infection is one of the most frequent and most lifesaving issues in neonatal care, especially for preterm infants. Newborn infants, especially premature ones, can change from a rather healthy to a severely ill status within hours. Therefore, quick but accurate decisions at the earliest signs of possible infection are paramount. Clinical symptoms, however, are nonspecific [1], and blood cultures are too slow to aid in the decision about initiating treatment. In order to avoid either unnecessary or untimely antibiotic use, a number of laboratory tests have been developed and evaluated for their use in the treatment decision following a sepsis workup [2,3,4]. Among these are measurements of serum proteins such as C-reactive protein (CRP), the interleukins IL-6 and IL-8, and procalcitonin [5,6,7,8,9].

In addition to protein measurements, changes in the relative proportions blood cellular components such as the proportion of immature and mature neutrophil granulocytes (I/T ratio) have been used. Most importantly, an increased I/T ratio (left shift) in a blood smear has been associated with a fairly good sensitivity for the detection of neonatal infection However, its specificity has been only moderate [10,11].

An important disadvantage of such cell differentiations is that they have to be carried out manually. A technician with a microscope has to differentiate the cells. As this is time-consuming, it is general practice to differentiate only 100 cells per blood smear. Since these 100 cells are picked randomly, statistical scatter is considerable and reduces diagnostic accuracy.

If the process of differentiating immature from mature neutrophils can be taken over by a machine, much higher numbers of cells can be counted, which can considerably reduce statistical scatter from cell selection. Furthermore, judgments concerning whether a cell is indeed immature or mature may also be more accurate if carried out by a machine, and automated blood cell analyses may be more readily accessible all around the clock [12,13].

With the introduction of the Sysmex 9000^®^ series (Sysmex Co., Kobe, Japan) machines [14,15] in our laboratory department, automated neutrophil differentiation became easily accessible for all clinical divisions of our University Hospital. However, literature data on the use of granulocyte differentiation by these machines are sparse. Recently, some data was published for the diagnosis of inflammation in children with chronic kidney disease [16]. Numbers are not directly comparable to manual differentiations since fewer cells are classified as immature by the machines. Neutrophils are only classified as immature up to the metamyelocyte stage. Band granulocytes are, unlike manual differentiations, classified as mature. Therefore, new algorithms for clinical decision-making based on automated neutrophil granulocyte differentiations are needed.

In this work, we set out to clarify how Sysmex 9000^®^ series granulocyte differentiations can help in the early diagnosis of neonatal infection. Furthermore, we evaluated their diagnostic utility in comparison to manual granulocyte differentiations. For this purpose, we extracted all sepsis workups from our hospital information system in which a manual and an automatic neutrophil granulocyte differentiation had been performed. Most importantly, we hypothesized that infants with confirmed infection had a higher proportion of immature neutrophils among all neutrophils, analogous to an increased I/T ratio in the manual counts.

## 2. Materials and Methods

### 2.1. Recruitment

This retrospective study was approved by the institutional review board of the medical faculty at the University of Leipzig, Germany. Infants were included if they were admitted to our neonatal unit, received at least one blood sampling with determination of serum CRP and complete blood count (including manual and automated neutrophil granulocyte differentiation), and did not fulfil any of the exclusion criteria. The latter were perinatal asphyxia, severe congenital heart disease, necrotizing enterocolitis, chromosomal anomalies, or surgical intervention. Multiple episodes of possible infection in the same infant were considered independent if at least 21 days had passed between them.

### 2.2. Group Assignment

Infants evaluated for sepsis were assigned into two groups according to the symptoms and blood result-based likelihood of infection. The “Unlikely Infection group” (UI) group consisted of infants who were very unlikely to have an infection based on the sepsis workup (IL-6 ≤ 400 pg/mL, CRP at the time of initial workup and 48 h later ≤ 10 mg/L, and negative blood culture results). In contrast, infants in the probable infection group (PI) were most likely to suffer an infection as indicated by at least one of the following sepsis indicators: at least one clinical symptom and CRP initially or the repeat measurement after 48 h ≥ 10 mg/L and/or IL-6 ≥ 400 pg/mL and/or positive blood culture. A subgroup of group PI is made up by the cases with positive blood culture (PC).

### 2.3. Blood Sampling for Cellular Analysis

Blood was sampled into K_3_EDTA vials and immediately sent to the laboratory. Blood smears were prepared by experienced laboratory technicians and stained by the May-Grünwald-Giemsa method. Cells were then differentiated by flow cytometry using Sysmex XN-9000^®^ (Sysmex Co., Kobe, Japan) series analyzers. Leukocytes were counted and differentiated using the WDF Channel. Immature granulocytes were determined using the DIFF mode (Xtra Vol.17.1). Quality controls were performed as recommended by the manufacturer and the Guideline of the German Medical Association (Bundesärztekammer).

### 2.4. Flow Cytometry

Automated cell differentiations (ACD) were performed by large-scale machines of the Sysmex XN-9000^®^ series. Blood samples were automatically aspirated, diluted and stained for nucleic acids. Thereafter, cells were moved to the flow chamber, where the cells were illuminated with a laser and scattered light was measured in all directions. Differences in light scatter were used to differentiate the cells (Xtra Vol. 17.1). Red blood cells were lysed before differentiating the white cells. The DIFF mode of the Sysmex XN-9000^®^ was used to differentiate the maturity of neutrophil granulocytes (Xtra Vol. 17.1). Cells up to the metamyelocyte stage were counted as immature, from the band stage onwards the cells were considered mature. This was distinctly different from the manual differentiations, where band cells were considered immature and only polymorphonuclear cells mature.

### 2.5. Xn-Ratio

Analogous to the I/T ratio determined from manual granulocyte differentiation, the xn-ratio was defined as the ratio of immature neutrophil granulocytes to the total neutrophil granulocytes as determined by automated differentiation.

### 2.6. Microscopic Differentiation

Experienced laboratory technicians differentiated the leucocytes using the CellaVision DM 96^®^ (Sysmex Co., Kobe, Japan) electronic microscope. Identified leucocytes were imaged automatically and displayed on a computer screen for differentiation until 100 leucocytes were completed. Band cells were differentiated from polymorphonuclear cells by the nuclear diameter at the narrowest part. If the latter was more than ⅓ of the largest nuclear diameter, it was a band cell, otherwise it was a polymorphonuclear cell.

### 2.7. Measurement of C-Reactive Protein Und Interleukin-6

The CRP concentrations were measured in samples of the patient ‘s blood serum by an immunologic particle-enhanced turbidimetric test on Cobas 6000^®^ c501 or Cobas 8000^®^ c701 machines (Roche Diagnostics, Mannheim, Germany). In this test, human CRP forms aggregates with latex particles covered with monoclonal anti-CRP antibodies (Cobas, V 7.0), thereby producing a measurable turbidity. IL-6 concentrations were measured by an electric chemiluminescence immunoassay (ECLIA) on Cobas 8000^®^ machines.

### 2.8. Data Handling and Statistics

Clinical and laboratory data were extracted from clinical databases, including electronic ICU charts, the database for external quality control, and the database of the laboratory system. Data was compared between groups generally by the Mann-Whitney U test, with a *p* value < 0.05 considered statistically significant. The diagnostic values of blood tests were determined by ROC analyses. The discriminatory power of the tests was considered low if the area under the curve (AUC) was <0.63, moderate if AUC was 0.63 up to <0.75, good if AUC was 0.75 up to <0.88 and excellent if the AUC was ≥0.88. SPSS Statistics^®^ software (Version 21.0) was used throughout.

## 3. Results

Records of 270 infants fulfilled inclusion criteria, but 103 of them also exclusion criteria. In 82 cases, insufficient blood sampling made it impossible to evaluate the cases for our study. A total of 9 cases were asphyxiated, 7 received surgery, and 5 had malformations. Thus, 167 infants were included (Table 1), who had 199 episodes of suspected infection, since 25 infants had 2 or more episodes. Of these, 101 episodes were classified, based on all diagnostic tests, as UI and 98 as PI. Positive blood cultures were found in 25 of the PI cases. Birth weight and gestational age were significantly lower in the PI group (*p* < 0.01).

### 3.1. Immature Granulocytes (Absolute Count)

The XN-9000^®^ determined absolute count of immature neutrophil granulocytes was significantly lower in the PI group than in the UI group (Figure 1). This effect was similarly seen in both subgroups when the infants in PI with and without a positive blood culture were analyzed separately. ROC analysis indicated that a cutoff of 0.265 × 10^9^/l was associated with a sensitivity of 80%, a specificity of 43%, a positive predictive value (PPV) of 58% and a negative predictive value (NPV) of 69% for the detection of a probable infection (Figure 2). We found similar results in the relative count of immature granulocytes (not shown).

### 3.2. Xn Ratio

A similar pattern was observed when using the proportion of immature neutrophil granulocytes among all neutrophil granulocytes, as determined by XN-9000^®^ (xn-ratio). The xn-ratio was significantly lower in PI infants (Figure 3). This effect was similarly seen in both subgroups when the infants in PI with and without a positive blood culture were analyzed separately. The ROC analysis showed a sensitivity of 80%, a specificity of 46%, PPV of 59%, and NPV of 70% at a cutoff value of 0.049 (Figure 4).

### 3.3. Immature to Total Ratio (Manual Complete Blood Cell Count)

Unlike what was known for the manually counted I/T ratio, XN-9000^®^ counts revealed lower numbers of immature granulocytes in the PI group. To determine whether our patient group behaved comparably to the literature, we also collected the manually counted I/T ratios. As expected, and unlike the machine results, manually counted I/T ratios were significantly higher in the PI group (Figure 5). This effect remained in both subgroups when the infants in PI with and without a positive blood culture were analyzed separately. The best discrimination was seen with a cutoff of 0.028, giving a sensitivity of 81%, specificity of 34%, PPV of 49%, and NPV of 68% for detecting a probable infection (Figure 6).

A separate analysis of subgroups of episodes at an age ≤72 h (early onset) and >72 h (late onset) or episodes in infants <32 weeks or ≥32 weeks did not yield different results (data not shown but available upon request).

## 4. Discussion

This is, to our knowledge, the first study evaluating the value of automated granulocyte differentiation by the Sysmex XN-9000^®^ system [13,17,18,19,20] for early diagnosis of neonatal sepsis. As such, it compared the diagnostic value of automated counts with that of manual counts. We succeeded in demonstrating that automated counts of immature granulocytes can be used to distinguish between PI and UI groups, with similarly moderate discriminatory power as manual counts. However, automated cell counts in neonates needed to be interpreted in a manner opposite from the usual processes when interpreting manual differentiations [14,16,21,22], since the absolute and relative counts of immature granulocytes were consistently and significantly lower in the PI group, while manual differentiations, as described before [10,11], yielded higher percentages of immature granulocytes in this group. This means, that an infection of a neonate is likely if the automatically counted proportion of immature granulocytes is lower (or the manually counted proportion is higher) than the set cutoff values.

Since both findings, lower automated counts of immature granulocytes and higher I/T ratios in manual counts were found in the same patients, and the results regarding the manual counts were similar to previous work [10,11]; unusual patient characteristics are unlikely the reason for the unexpected results with the automated counts. Instead, methodological differences in cell differentiation must be considered. In manual counts, neutrophil granulocytes are differentiated by morphology, most importantly by the shape of the nucleus. Granulocyte precursor cells up to the metamyelocyte stage, and band cells are considered immature. Band cells (the last stage of immaturity) are differentiated from polymorphonuclear (mature) granulocytes by the nuclear diameter at the narrowest part. In contrast, the machine differentiates cells by the scatter of laser beams at internal cellular structures. This methodology results in distinctly different results in classifying neutrophil granulocytes in comparison to the manual complete blood cell counts, and, most importantly, in the assignment of band cells. Band cells are classified as mature neutrophil granulocytes in automated differential blood cell counts. Since band cells are the most abundant immature cells in manual counts, this makes a large difference. The proportion of immature cells in automated counts is lower than in manual counts by a factor of almost 10.

Our results show, that in case of oncoming infection, the proportion of manually counted immature cells (which include band cells) increases, while the proportion of automatically counted immature cells, which do not include band cells, decreases. This observation may indicate that an oncoming neonatal infection increases band cells while depleting the more immature precursors. This would be a new finding, since the proportions of different stages of granulocyte precursors in oncoming neonatal infection have not yet been systematically investigated. Certainly, this hypothesis requires additional studies for confirmation.

Differing results were found with Sysmex XE-2100^®^, a previous series of automated blood cell counters [20]. However, it is unclear at present whether Sysmex XE-2100^®^ and XN-9000^®^ use the same criteria for categorizing cells into immature and mature. In Sysmex XE-2100^®^, the “IG” counts were only displayed on the research screen and were not available for clinical routine. Since these machines are more than 20 years old, changes in algorithms are likely. Unfortunately, there is no information on such algorithms and their revisions available from the manufacturer. Furthermore, differences in the timing between onset of symptoms and blood and blood culture sampling between our study and the one by Nigro et al. [20] may have played a part. Finally, only infants with positive blood culture were considered as having an infection, which may have missed part of the infected population since the positive blood culture yield is generally low, also in the study of Nigro et al., where positive blood cultures only occurred in infants >seven days of age. [20] Numbers of infants with an elevated CRP or antibiotic treatment were not reported.

With the low yield of positive blood cultures in neonatal infections, many infected infants are falsely classified as uninfected if the classification is only based on positive blood cultures. Previous studies have shown that elevated CRP and interleukins have a high sensitivity and specificity [2,3,6,10,23], and repeated CRP has also been shown to be a good parameter for confirming infection as well as adjusting treatment duration [24]. A low CRP after 48 h is also a great help to end antibiotics early if a suspected infection was not real. Since other causes of CRP elevations (such as autoimmune diseases) are very rare in the neonatal period, we think that classifying infants based on repeated CRP measurements as well as blood cultures brings a better distinction between infected and uninfected cases.

### Limitations of This Study

Owing to the retrospective design of this study, some confounding effects may persist. We had to select patients who received automated and manual cell differentiations simultaneously. As there were no rules concerning which types of blood counts were to be ordered in clinical routine, the orders depended on the habits of different physicians on duty and not on characteristics of the patients. Therefore, systematic differences between infants receiving automatic, manual, both, or no leucocyte differentiations during the sepsis workup are unlikely. Furthermore, indications for undertaking a sepsis workup were not completely standardized, and therefore the time interval from onset of infection until the decision to take a blood sample was variable. The technicians doing the manual differentiations were not specifically selected, and all trained technicians in our lab participated. However, the results of the manual differentiations were as expected, which rules out systematic errors origination from patient inclusion as well as technician participation.

Our sample size was limited, since we had to limit our patient group to cases with simultaneous determinations of automated and manual differentiations. Thus, we had insufficient numbers to distinguish between early- (age <72 h) and late-onset (≥72 h) infections. Preliminary analyses however, revealed that these subgroups did not show trends that were grossly different from the whole group, and indicated similar cutoff values.

Furthermore, groups appear unbalanced regarding the timing of sepsis workups. Early sepsis workups in the first 72 h were more often negative, resulting in UI classification. The threshold for ordering a sepsis workup is probably lower in the first 72 h, since respiratory distress syndrome is more frequent, which may mimic infection and thus induce more sepsis workups.

Some imbalances in the patient group may also introduce some bias [4]. Sepsis workups were more often negative in the first three days of life, resulting in a higher proportion of early sepsis workups in the UI group, and more late sepsis workups in the PI group. Subgroup analyses, although hampered by low group sizes, showed no deviant behavior of any subgroup from the whole patient collective, which makes a strong bias from this effect unlikely.

Overall, possible confounding effects do not appear strong enough to invalidate the results. While the current results are unexpected in face of the established increased frequency of immature granulocyte precursors in manual blood counts, these results demonstrate that an automated blood count by FACS using the Sysmex XN-9000^®^ analyzer detects a signal indicating an emerging severe infection. Unfortunately, the exact algorithm used by the Sysmex XN-9000^®^ analyzer is currently not available, precluding a precise characterization of the signal detected. Independently of the analyzer used, the current results call for future studies using alternative approaches for blood count analyses, which may reveal algorithms allowing faster and more reliable detection of an emerging severe infection.

It is unknown whether it is possible with the Sysmex XN-9000^®^, to detect band cells separately and assign them to the immature fraction. The differences in laser light scatter between band and polymorphnuclear cells may be too small. If such a software change is implemented, our deductions may no longer be valid.

## 5. Conclusions

We conclude that a low count of immature granulocytes and a low xn-ratio as determined by Sysmex XN-9000^®^ may indicate a neonatal infection. The known increase of the I/T ratio, which is mainly dependent on band cells, appears to be accompanied by a depletion of more immature precursors, which constitute the cells classified as immature by the XN-9000^®^. As a result, low counts of immature neutrophil granulocytes and low xn-ratios in the automatic counts indicate infection, as well as increased I/T ratios in manual counts. A low count of immature granulocyte precursors in an automated, FACS based blood count should therefore not be mistaken as a low risk of infection, but rather raise suspicion of an emerging severe infection. Clinically useful cutoff values for detection of neonatal infections may be <0.265 for the automatic immature granulocyte count, <0.049 for the xn-ratio, and >0.028 for the manual I/T ratio, all of which have a sensitivity of >80%. However, a single laboratory test is not sufficient to rule out or confirm a neonatal infection. In addition to clinical findings, several different laboratory tests, such as cell counts, acute phase reactants and cytokine measurements, are needed [1]. Costs for Sysmex XN-9000^®^ counts are probably lower than manual counts since less technician time is needed.

## Figures and Tables

**Figure 1 jcm-11-05507-f001:**
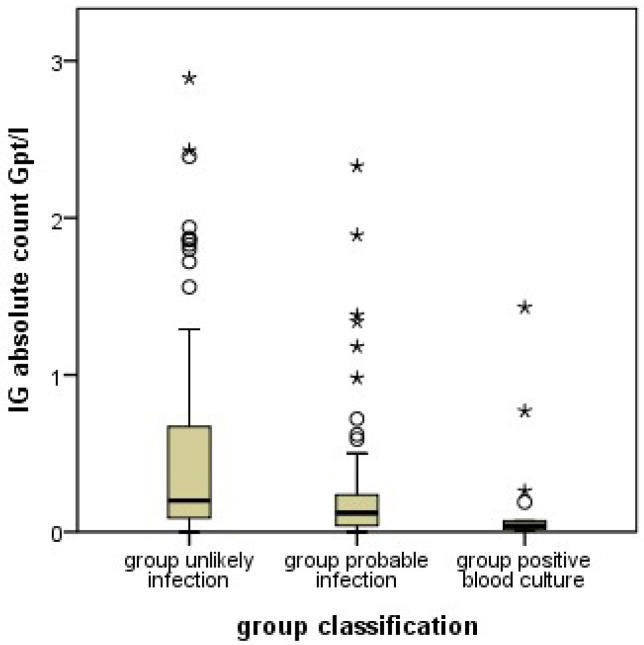
Boxplot of the numbers of immature granulocytes (IG) as counted by the Sysmex XN-9000^®^ flow cytometers in the unlikely infection (UI), probable infection (PI) groups as well as the positive blood culture (PC) subgroup. The PI group had lower counts of immature granulocytes than the UI group (*p* < 0.001). This effect was even stronger in the PC subgroup with positive blood culture. The boxes show median, and interquartile range, the whiskers 95% confidence range, ○ = outliers, ★ = extreme outliers.

**Figure 2 jcm-11-05507-f002:**
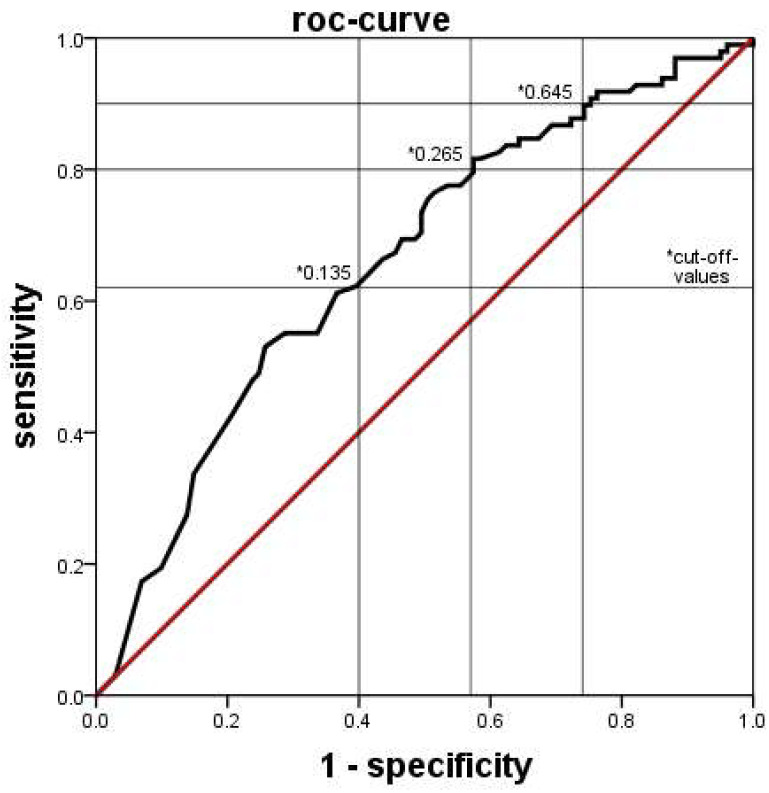
ROC-curve of the diagnostic accuracy of detecting a probable infection by the numbers immature granulocytes as counted by the Sysmex XN-9000^®^. The area under the curve was 0.661. * Several different cut-off values with their respective sensitivity and specificity are indicated.

**Figure 3 jcm-11-05507-f003:**
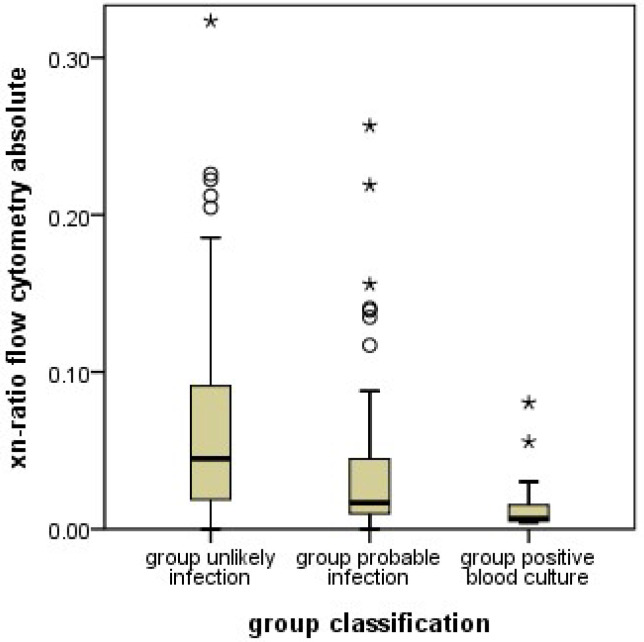
Boxplot of the xn-ratios (number of immature granulocytes divided by the total number of granulocytes) as counted by the Sysmex XN-9000^®^ flow cytometers in the unlikely infection (UI), probable infection (PI) groups, and the positive blood culture (PC) subgroup. The PI group had lower xn-ratios than the UI group (*p* < 0.001). This effect was even stronger in the PC subgroup of PI with positive blood culture. The boxes show median, and interquartile range, the whiskers 95% confidence range, ○ = outliers, ★ = extreme outliers.

**Figure 4 jcm-11-05507-f004:**
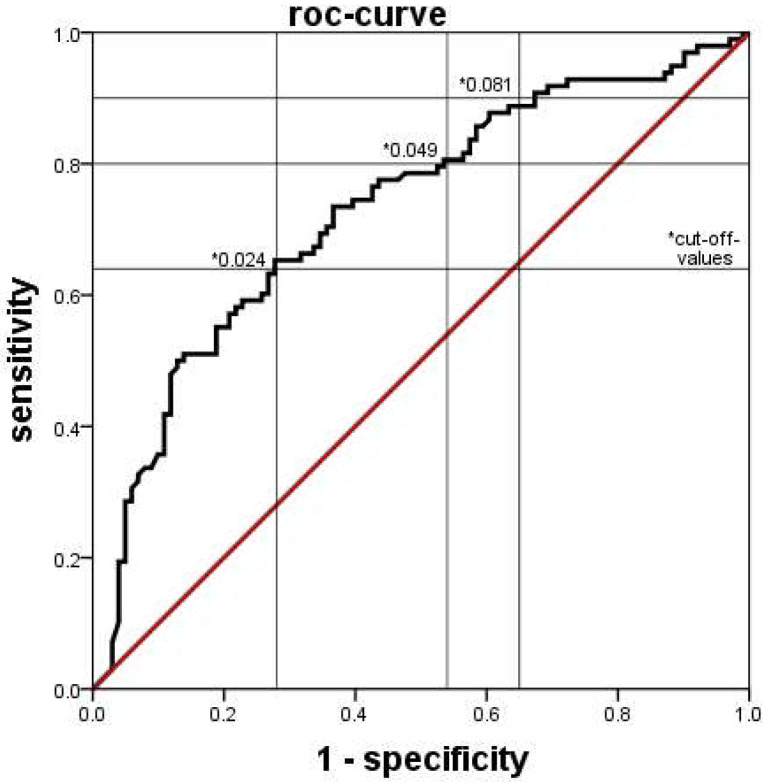
ROC-curve of the diagnostic accuracy of detecting a probable infection by the xn-ratio as determined by the Sysmex XN-9000^®^. The area under the curve was 0.727. * Several different cut-off values with their respective sensitivity and specificity are indicated.

**Figure 5 jcm-11-05507-f005:**
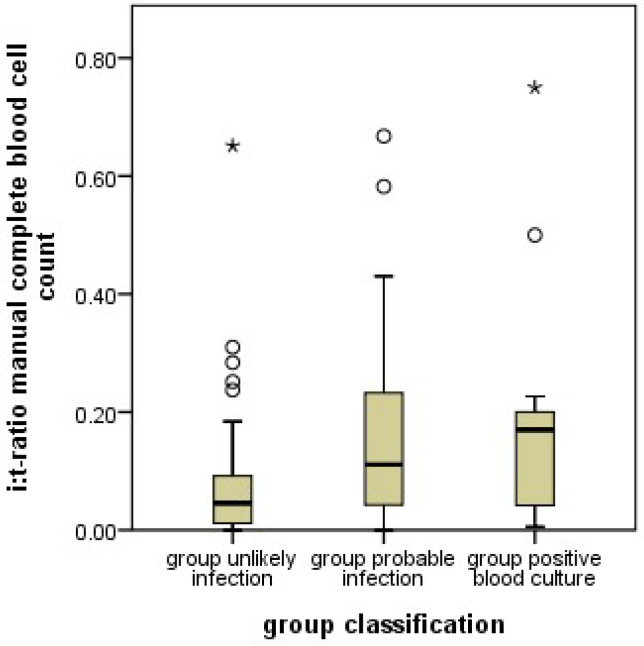
Boxplot of the I/T-ratios (number of manually counted immature granulocytes divided by the total number of granulocytes) in the unlikely infection (UI), probable infection (PI) groups, and the positive blood culture (PC) subgroup. The PI group had higher I/T-ratios than the UI group (*p* < 0.001). The boxes show median, and interquartile range, the whiskers 95% confidence range, ○ = outliers, ★ = extreme outliers.

**Figure 6 jcm-11-05507-f006:**
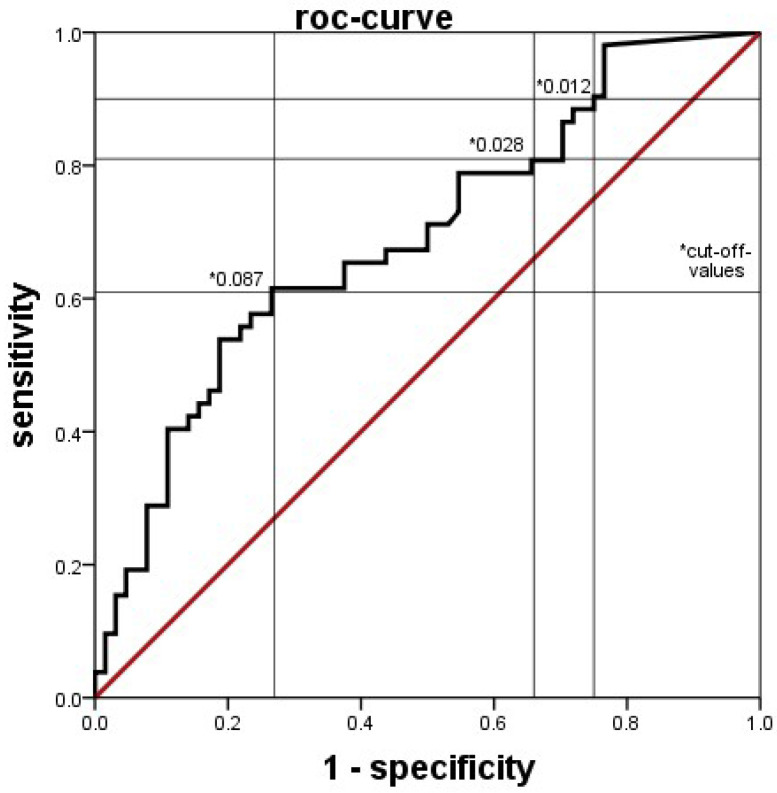
ROC-curve of the diagnostic accuracy of detecting a probable infection by the manually counted I/T-ratio. The area under the curve was 0.692. * Several different cut-off values with their respective sensitivity and specificity are indicated.

**Table 1 jcm-11-05507-t001:** Demographic characteristics of the study groups (GA = gestational age).

	Group Unlikely Infection	Group Probable Infection	Group Positive Blood Culture
Male (GA < 37 weeks)	63 (43)	60 (45)	13 (9)
Female (GA < 37 weeks)	38 (29)	38 (30)	12 (10)
Median (min-max) gestational age in weeks	32 4/7 (23 4/7–41 6/7)	28 3/7 (23 4/7–42 0/7)	28 4/7 (24 3/7–39 4/7)
median (min-max) birth weight in g	1950 (398–4200)	940 (375–6110)	1042 (640–3906)
median (average) age at time of blood sampling (days)	0 (3)	10 (22)	11 (20)

## Data Availability

The data set is pseudonymized, but this may not completely exclude identification of individual patients. Therefore, in accordance with our IRB, the full data set cannot be made publicly available. Excerpts of the data set can be made available to individual researchers upon request.

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
