# Peer review of "Automated Complete Blood Cell Count Using Sysmex XN-9000® in the Diagnosis of Newborn Infection"

_jcm, 2022, doi:10.3390/jcm11195507_

Round 1
Reviewer 1 Report
1. We conclude that a low count of immature granulocytes as determined by Sysmex XN-9000 may indicate neonatal infection.
So, if the soft ware is adjusted to detect band cells as a immature granulocytes, this type of deduction may not arise.
So classical I/T ratio > 2 indicative of PI hold true instead of this opposite finding which confuse among many paediatricians.
2. The diagnostic accuracy of Sysmex XN-9000, the area under the curve was 0.727. ROC-curve of the diagnostic accuracy of detecting a probable infection by the manually counted I/T-ratio, the area under the curve was 0.692. So, there is a marginal difference. How to justify the use Sysmex XN-9000, in terms of cost effectiveness. Also, single parameter is never used in diagnosing and treating the neonatal sepsis. Neonal sepsis diagnosis criteria laboratory and clinical https://www.nature.com/articles/s41390-020-0785-x/tables/3 & https://www.nature.com/articles/s41390-020-0785-x/tables/4
3. Is this interpretation hold true for ELBW, VLBW babies, immunocompromise babies ?
4. Vast limitation of this study with many confounding factor, applicability of this conclusion in general is a question mark.
Author Response
"We conclude that a low count of immature granulocytes as determined by Sysmex XN-9000® may indicate neonatal infection."
Response: Yes, this is our main message.
"So, if the soft ware is adjusted to detect band cells as a immature granulocytes, this type of deduction may not arise."
Response: The reviewer raises a good point. If such a software change is implemented, our deductions may no longer be valid. However, it is unknown whether it is possible with the Sysmex XN-9000®, to detect band cells separately and assign them to the immature fraction. The differences in laser light scatter between band and polymorphnuclear cells may be too small. We have added this point to the discussion.
"So classical I/T ratio > 2 indicative of PI hold true instead of this opposite finding which confuse among many paediatricians."
Response: The reviewer is correct, we found the classical I/T ratio to behave like ever before in our patients.
"2.The diagnostic accuracy of Sysmex XN-9000, the area under the curve was 0.727. ROC-curve of the diagnostic accuracy of detecting a probable infection by the manually counted I/T-ratio, the area under the curve was 0.692. So, there is a marginal difference. How to justify the use Sysmex XN-9000, in terms of cost effectiveness. Also, single parameter is never used in diagnosing and treating the neonatal sepsis. Neonal sepsis diagnosis criteria laboratory and clinical https://www.nature.com/articles/s41390-020-0785-x/tables/3 & https://www.nature.com/articles/s41390-020-0785-x/tables/4"
Response: The reviewer is correct. In terms of cost effectiveness, the SYSMEX XN-9000® is superior because the most expensive component of a lab is personnel, and manual counts require a lab technician to be performed. We have added this information.
Furthermore, the reviewer is correct that multiple criteria are necessary, which we wrote in the final paragraph. We have added the suggested citation.
"3. Is this interpretation hold true for ELBW, VLBW babies, immunocompromise babies ?"
Response: Yes, because all named patient types except immune deficient babies were included in the study.
"4. Vast limitation of this study with many confounding factor, applicability of this conclusion in general is a question mark."
Response: Yes, we discuss the limitations of the study and have expanded this setion.
Author Response
In this article it is suggested that automatically counted immature granulocytes can probably be used to distinguish between infected and uninfected neonates.
Comments
I think that this study has more limitations than those that the authors describe at the end of the discussion section.
- Line 67: “we hypothesized that infants with confirmed infection”. How many of the neonates/infants enrolled had a culture-proven infection?
Response: We have added this information to the manuscript.
- Group assignment section. Conventional culture techniques remain the “gold standard” to confirm the diagnosis of neonatal sepsis. (1) Most of the studies, referring to the diagnosis of neonatal sepsis using inflammatory biomarkers divide patients in groups of a) patients with proven sepsis and b) patients with suspected or clinical sepsis. I believe this should be performed in this study, as well.
Response: We have included the data on the subgroup of infants with positive blood cultures. The behaviour of cell counts in biomarker confirmed infection and culture confirmed infection was similar.
- Line 138: “Records of 270 infants fulfilled inclusion criteria, but 103 of them also exclusion crιteria”. The amount of the excluded infants reported by the authors stands for approximately 40% of the total amount of infants, that is rather high. Please, provide extra information on the causes of exclusion or provide a patients’ flowchart.
Response: We have included this information in the manuscript
- Line 165: I would like to see the early/late onset sepsis data, please!
Respnse: We provide the data here as follows:
|
Absolute IG count, EOS |
number |
Median |
Minimum |
Maximum |
|
UI |
91 |
0,23 |
0,00 |
4,80 |
|
PI |
39 |
0,14 |
0,00 |
2,33 |
|
Absolute IG count, LOS |
number |
Median |
Minimum |
Maximum |
|
UI |
10 |
0,11 |
0,05 |
0,39 |
|
PI |
59 |
0,05 |
0,00 |
6,37 |
|
Xn ratio, EOS |
number |
Median |
Minimum |
Maximum |
|
UI |
91 |
0,05 |
0,00 |
0,32 |
|
PI |
39 |
0,03 |
0,00 |
0,26 |
|
Xn ratio, LOS |
number |
Median |
Minimum |
Maximum |
|
UI |
10 |
0,02 |
0,01 |
0,05 |
|
PI |
59 |
0,01 |
0,00 |
0,22 |
|
I/T ratio, EOS |
number |
Median |
Minimum |
Maximum |
|
UI |
61 |
0,05 |
0,00 |
0,65 |
|
PI |
25 |
0,12 |
0,00 |
0,75 |
|
I/T ratio, LOS |
number |
Median |
Minimum |
Maximum |
|
UI |
3 |
0.03 |
0.0 |
0.16 |
|
PI |
27 |
0.1 |
0.006 |
0.67 |
- In Table 1, one infant with PI weighs 6.110 gr. What day of life was it in?
Response: This infant was 3 days old
- I/T ratio varies with postnatal age (1). The authors in line 269 state that ”Early sepsis workups in the first 72 hours were more often negative, resulting in UI classification”. Do the authors think that this would possibly bias the study, taking into account that the UI group consists mostly of babies in the early days of life?
Response: Generally, an infant is more likely to get a sepsis workup shortly after birth, because of the similarity of some symptoms in RDS and infection. This results in more negative sepsis workups in the early days. Low sample sizes in the < 72 and > 72 hours subgroups however, prevent meaningful subgroup analyses. We performed these analyses anyway and found that all tended in the same direction, similar to the overall analyses presented in the paper. There was no subgroup running totally different. We have included this important aspect in the discussion of limitations.
- Figures 4,6: The discriminatory power of automated vs manually counted xn-I/T ratio is moderate, according to the authors.
Response: Yes, this is correct. We have added this to the discussion.
- Lines 273-274: What is the policy of the NICU regarding the evaluation of neonates? Is it performed solely by nurses?
Response: The neonates are regularly evaluated by nurses and physicians. Both work in shifts (no night calls with bed time) 24/7. The sentence was misleading and thus removed.
- The discussion section is quite weak. A major limitation of this study is that it is retrospective, with a limited sample size. This is a study that compares effectively the two methods, automated and manually counted granulocytes, rather than a study comparing two methods to predict neonatal sepsis.
Response: The reviewer is correct. The discussion was expanded and includes these points.
References:
- Istemi Han Celik,, Morcos Hanna,, Fuat Emre Canpolat,, Mohan Pammi. Diagnosis of Neonatal Sepsis: The Past, Present and Future. Pediatr Res. 2022 January ; 91(2): 337–350. doi:10.1038/s41390-021-01696-z.
Response: We have added this reference

Reviewer 3 Report
This study retrospectively investigated the value of automated cell differentiation system (Sysmex XN-9000® 2) in the detection of neonatal infection. The authors found the absolute count of immature granulocytes was significantly lower in the group of probable infection (PI) compared to unlikely infection (UI) neonates. However, manually counted immature to total (I/T) ratios of granulocytes were higher in PI than in UI infants.
1. The major drawback of this study is the definition of neonatal infections. CRP, IL-6, or any inflammatory markers are not reliable parameters to differentiate between infection and inflammation. Moreover, the authors did not report the ratio of patients with positive blood culture or pathogens identification. The etiology of infection/inflammation may further bias the study result.
2. Similar study (PMID: 15743752) using automated cell differentiation system (Sysmex XE-2100) showed elevated immature granulocytes counts by manual and automated methods were associated significantly with positive blood culture results. This study used blood culture positive result compared to culture negative case, which is a more convincing definition for neonatal infection. More in depth discussions are encouraged based on study design and case definition.
3. In table 1, some data in the row of “Median (min-max) gestational age in weeks” are missing.
4. Some typing errors need to be corrected. For example, imature in line 155, absolut in line 143.
Author Response
"This study retrospectively investigated the value of automated cell differentiation system (Sysmex XN-9000® 2) in the detection of neonatal infection. The authors found the absolute count of immature granulocytes was significantly lower in the group of probable infection (PI) compared to unlikely infection (UI) neonates. However, manually counted immature to total (I/T) ratios of granulocytes were higher in PI than in UI infants."
1."The major drawback of this study is the definition of neonatal infections. CRP, IL-6, or any inflammatory markers are not reliable parameters to differentiate between infection and inflammation. Moreover, the authors did not report the ratio of patients with positive blood culture or pathogens identification. The etiology of infection/inflammation may further bias the study result."
Response: This is an important point which atracts controversies for moree than 20 years. With the low yield of positive blood cultures in neonatal infections, many infected infants are falsely classified as uninfected if the classification is only based on positive blood cultures. Previous studies have shown that elevated CRP and interleukins have a high sensitivity and specificity, and repeated CRP has also been shown to be a good parameter for confirming infection as well as adjusting treatment duration. A low CRP after 48 hours is also a great help to end antibiotics early if a suspected infection was not real. Since other causes of CRP elevations, such as autoimmune diseases, are very rare in the neonatal period, we think that classifying infants based on repeated CRP measurements as well as blood cultures brings a better distinction between infected and uninfected cases.
We have added this paragraph to the discussion. Furthermore, we have added the reuslts of infection episodes with positive blood culture as sparate colums in figs 1,3, and 5, and show that cell counts tended in the same direction, regardless whether the infection was confirmed by CRP only or by positive blood culture.
2."Similar study (PMID: 15743752) using automated cell differentiation system (Sysmex XE-2100) showed elevated immature granulocytes counts by manual and automated methods were associated significantly with positive blood culture results. This study used blood culture positive result compared to culture negative case, which is a more convincing definition for neonatal infection. More in depth discussions are encouraged based on study design and case definition."
Response: This is another imprtant point. It is true that the older publication on Sysmex XE-2100® shows results which are opposite to ours. However, it is unclear at present whether Sysmex XE-2100® and XN-9000® use the same criteria for categorizing cells into immature and mature. In Sysmex XE-2100®, the “IG” counts were only displayed on the research screen and were not available for clinical routine. Since these machines are more than 20 years old, changes in algorithms are likely, and my be the reason for the differing results. Unfortunately, there is no information on such algorithms and their revisions available from the manufacturer. Furthermore, differences in the timing between onset of symptoms and blood and blood culture sampling between our study and the one by Nigro et al. may have played a part. Finally, only infants with positive blood cul-ture were considered as having an infection, which may have missed part of the infected population since the positive blood culture yield is generally low, also in the study of Nigro et al., where positive blood cultures only occurred in infants >7 days of age. Numbers of infants with an elevated CRP or antibiotic treatment were not reported.
We have added this information to the discussion
3. "In table 1, some data in the row of “Median (min-max) gestational age in weeks” are missing."
Response: We are sorry for the mistake and added the data.
4. "Some typing errors need to be corrected. For example, imature in line 155, absolut in line 143."
Response: We thank for the thorough review and corrected all typos we could find.
Round 2
Reviewer 1 Report
Nicely edited. Conclusion should be precise.
Author Response
Thank you for this advice We have reworded parts of the conclusions in the Abstract and at the end of the manuscript accordingly.
Reviewer 3 Report
The authors have addressed all the issues I mentioned. I have no further comment.
Author Response
Thank you for aur advice